# *Bacillus subtilis* and *Rhizophagus intraradices* Improve Vegetative Growth, Yield, and Fruit Quality of *Fragaria × ananassa* var. San Andreas

**DOI:** 10.3390/microorganisms12091816

**Published:** 2024-09-02

**Authors:** Lucero Huasasquiche, Leonela Alejandro, Thania Ccori, Héctor Cántaro-Segura, Tomás Samaniego, Kenyi Quispe, Richard Solórzano

**Affiliations:** 1Estación Experimental Agraria Donoso, Dirección de Desarrollo Tecnológico Agrario, Instituto Nacional de Innovación Agraria (INIA), Lima 15200, Peru; lucero.26.lhs@gmail.com; 2Facultad de Agronomía, Universidad Nacional Agraria La Molina (UNALM), Lima 15024, Peru; leonelak2@gmail.com (L.A.); thaniaccori@gmail.com (T.C.); hcantaro@lamolina.edu.pe (H.C.-S.); 3Estación Experimental Agraria Donoso, Dirección de Supervisión y Monitoreo de las Estaciones Experimentales, Instituto Nacional de Innovación Agraria (INIA), Lima 15200, Peru; td.samaniego@gmail.com; 4Centro Experimental La Molina, Dirección de Supervisión y Monitoreo de las Estaciones Experimentales, Instituto Nacional de Innovación Agraria (INIA), Lima 15024, Peru; k.quispe1008@gmail.com; 5Facultad de Ciencias Ambientales, Universidad Científica del Sur (UCSUR), Lima 15067, Peru

**Keywords:** fruit quality, microbial inoculants, mycorrhizae, strawberry growth, yield

## Abstract

Strawberry cultivation requires strategies that maintain or improve its yield within a scheme in which reducing fertilizers and other chemical products can make its consumption safer and more environmentally friendly. This study aims to evaluate the effect of *Bacillus subtilis* and *Rhizophagus intraradices* on strawberry growth, yield, and fruit quality. *B. subtilis* and *R. intraradices* were inoculated and co-inoculated under three fertilization levels of 225-100-250, 112-50-125, and 0-0-0 kg∙ha^−1^ of N, P_2_O_5_ and K_2_O. Vegetative growth was evaluated in plant height (cm), leaf area (cm^2^), aerial fresh weight (g), aerial dry weight (g), and plant coverage (%) variables. Fruit quality parameters such as total acidity (g∙100 mL^−1^), soluble solids (Brix°), and firmness (kg) were also determined, as well as the number of fruits per m^2^ and yield (t∙ha^−1^). The results showed that the pre-treatment of root immersion in a nutrient solution with *B. subtilis* and the fractionation of 6 L *B. subtilis* inoculation per plant at a concentration of 10^7^ CFU∙mL^−1^, in combination with 225-100-250 kg∙ha^−1^ of N, P_2_O_5,_ and K_2_O, achieved the highest accumulation of dry matter (12.9 ± 1.9 g∙plant^−1^), the highest number of fruits (28.2 ± 4.5 fruits∙m^−2^), and the highest yield (7.2 ± 1.4 t∙ha^−1^). In addition, this treatment increased the soluble sugar content by 34.78% and fruit firmness by 26.54% compared to the control without inoculation. This study highlights the synergistic effect of mineral nutrition and microbial inoculation with *B. subtilis* in increasing strawberry yield and fruit quality.

## 1. Introduction

Nowadays, strawberry (*Fragaria* sp.) has been reported to be a popular fruit crop that has garnered significant interest over the past several decades, leading to increased economic demand for specialty crops [1]. Based on FAO reports, the worldwide production of strawberries exceeded 9.18 million tonnes in 2021. Asia is the largest continental producer (4.53 million tonnes) followed by America (2.18 million tonnes), Europe (1.76 million tonnes), Africa (634.29 thousand tonnes), and Oceania (60.31 thousand tonnes). In 2020, the global strawberry production value reached USD 14 billion [2], which is consistent with its increasing demand.

Mineral fertilizers have been a key solution to the world’s high food demand for many years. Following the Second World War, synthetic or chemical fertilizers rapidly increased agricultural productivity, meeting the needs of the growing global population [3]. These inputs gained prominence as low-cost, easy-to-handle, and quick sources of nutrients for crops [3]. However, they primarily benefit the plants while neglecting the health of surrounding resources, such as soil, water, and air.

Several studies show that the excessive use of chemical fertilizers leads to the long-term accumulation of heavy metals like cadmium, copper, zinc, and lead in the soil [4,5,6]. High application rates or improper use of chemical fertilizers can exacerbate the accumulation of heavy metals in the soil [7]. Over time, the repeated application of fertilizers containing heavy metals can lead to elevated concentrations in the soil. These heavy metals can be mobilized through soil erosion and surface runoff, potentially spreading beyond the fertilized area and contaminating water systems, which can indirectly affect soil health. Plants can absorb heavy metals from contaminated soil, and if soil contamination occurs due to fertilization practices, these metals may enter the food chain, posing risks to human health and ecosystems. Additionally, they contribute to phosphate and nitrate pollution [8,9], eutrophication [10], and alter the composition of microbial populations [11], among other issues, causing a negative impact on the environment and consequently on human health [12].

Microbial inoculation has emerged as a sustainable alternative to the excessive use of synthetic fertilizers. This technology involves adding microorganisms with plant growth-promoting activities to the soil or directly to the seed [13]. These microorganisms can produce non-volatile compounds (e.g., phytohormones, siderophores) or volatile organic compounds (VOCs) that diffuse between soil particles. Both have beneficial effects on the plant, either by promoting its growth, inducing systemic resistance, or protecting it against abiotic stress factors [14] or pathogens [15]. In addition, they can also improve the efficiency of other organic technologies such as hydroponics [16], which is increasingly being applied in strawberry cultivation. Despite the reported advantages in various crops, microbial inoculation remains a technique that still needs to be explored compared to the extensive agricultural practices worldwide [17].

In Peru, there are several products based on beneficial microorganisms. However, the market is still developing, and microbial inoculation has not been adopted as part of integrated agronomic management. One of the main reasons for this slow adoption is the lack of dissemination and field studies to evaluate its efficiency across different crops and under various conditions. Factors such as crop, soil conditions, inoculant quality, and agricultural practices must be considered when applying beneficial microorganisms [18]. For example, it is known that the use of chemical fertilizers has an influence on microbial populations in the soil [19], and therefore it is necessary to study the combined effect of these agrochemicals with microbial inoculants, to understand and overcome barriers that may reduce their effectiveness.

The present study evaluated the effect of inoculation of *Bacillus subtilis* and *Rhizophagus intraradices* on the growth, yield and quality of the strawberry crop (*Fragaria* × *ananassa* cv. San Andreas), under three mineral fertilization scenarios: 225-100-250 (full dose), 112-50-125 (half dose), and 0-0-0 (without fertilization) kg∙ha^−1^ of N, P_2_O_5_ and K_2_O. All of this was performed with the objective of analyzing the influence that mineral fertilization has on the efficiency of microbial inoculation. Given that, in the current context, we cannot become totally independent of agrochemicals, it becomes important to evaluate their effect on sustainable technologies such as microbial inoculation.

## 2. Materials and Methods

### 2.1. Experimental Area Characteristics

Bare-root strawberry (*Fragaria* sp. Var. San Andreas) seedlings, three months old, were obtained from the Programa Nacional de Investigación en Hortalizas. Transplanting was carried out in a 172.8 m^2^ field, with a distance of 0.8 m between furrows and 0.2 m between plants. Watering was performed 1 to 2 times per week and manual weeding was carried out every two weeks.

The experiment was conducted between June and December 2023, at the Donoso Agrarian Experiment Station of the National Institute for Agrarian Innovation (INIA), in Huaral. During the experiment, the average temperature was 19.5 °C, the average maximum temperature was 23.3 °C, the average minimum temperature was 17.0 °C, and the relative humidity was 80.5%.

The soil in the experimental field consists of 61% sand, 14% silt, and 25% clay, classifying it as sandy clay loam in texture [20]. The soil pH was 8.0 [21], with an electrical conductivity of 0.11 dS∙m^−1^ [22]. Available phosphorus was 50.41 mg∙kg^−1^, and available potassium was 115.89 mg∙kg^−1^. The organic matter content was 0.7%, with available iron at 3.17 mg∙kg^−1^, available zinc at 3.66 mg∙kg^−1^, available copper at 2.76 mg∙kg^−1^, and available manganese at 4.07 mg∙kg^−1^ [20]. The calcium carbonate (CaCO_3_) content was 19.20% [20]. Cationic relationships in the soil indicated concentrations of 7.07 mEq∙L^−1^ for Ca^2+^, 2.70 mEq∙L^−1^ for Mg^2+^, 0.34 mEq∙L^−1^ for Na^+^, and 0.3 mEq∙L^−1^ for K^+^, with an effective cation exchange capacity (CEC) of 10.41 mEq∙L^−1^. These characteristics qualify the soil as deficient in potassium, due to a low percentage of exchangeable potassium at 2.88%, resulting in a low level of available potassium. Additionally, the soil has a low fertility potential due to a shallow organic matter content and a pH of 8, limiting micronutrient availability.

### 2.2. Experimental Design and Treatments

The experimental plot was set up using a bifactorial block design. The first factor was fertilization at three levels: 225-100-250 (full dose), 112-50-125 (half dose), and 0-0-0 kg∙ha^−1^ (without fertilization) of N, P_2_O_5_, and K_2_O, respectively. The second factor was microbial inoculation with four levels: individual inoculation with *Bacillus subtilis* and *Rhizophagus intraradices*, co-inoculation with both microorganisms, and a control with no inoculation. Considering both factors, a total of 12 treatments were evaluated, each with 3 repetitions (Table 1). Each repetition covered an individual experimental area of 4.8 m^2^.

### 2.3. Microbial Inoculation

The *Bacillus subtilis* strain used in this research was isolated by Solórzano-Acosta and Quispe [23] from avocado rhizospheric soil. The preparation of the inoculum began with the incubation of the microorganism in a nutrient solution composed of 0.5% peptone, 0.3% yeast extract, and 0.5% sodium chloride, at a 6.8 pH, for 3 days at 28 °C, until reaching a concentration of 10^9^ CFU∙mL^−1^ (Solution A). Subsequently, solution A was diluted in non-chlorinated water at a 10 mL∙L^−1^ ratio, obtaining a solution with a final concentration of 10^7^ CFU∙mL^−1^ (Solution B, inoculant).

The inoculation with *Bacillus subtilis* involved applying Solution B four times throughout the trial: at transplanting, and at 22, 51, and 78 days after transplanting (dat). The first inoculation was performed by immersing the roots in Solution B for 15 min immediately before transplanting [24]. The three subsequent inoculations were carried out by adding 2 L of inoculant per plant via drench application.

For mycorrhizal inoculation, a *Rhizophagus intraradices* strain isolated by Castañeda et al. [25] from the rhizosphere of *Sporobolus* sp., predominant in a wetland in Ica, Peru, was used. The inoculum consisted of a mixture of rootlets colonized 70–80% by the fungus and spores (1700 spores per gram) mixed in a carrier based on biochar and sand. The mycorrhizal inoculation was carried out only at transplanting through direct contact with the moist roots, applying approximately 0.5 g of inoculum per plant. For co-inoculated treatments, the roots were immersed in the *B. subtilis* suspension (Solution B), and then 0.5 g of mycorrhizal inoculum was added per plant when the roots were wet, thus facilitating adherence to them.

### 2.4. Fertilization Treatments

Fertilization was divided into four stages, occurring at 4, 36, 51, and 63 days after transplanting (dat). The amounts were calculated based on a previous soil analysis of the experimental field and as recommended by Olivera [26]. In all treatments, an initial fertilization of 30-60-30 kg∙ha^−1^ of N (urea), P_2_O_5_ (potassium sulfate), and K_2_O (diammonium phosphate) was applied at 4 dat. Table 2 shows the fertilizers used and the amounts applied on the four fertilization dates for the full fertilization treatments (F100) and with 50% of the full dose (F50).

### 2.5. Biometric Parameters

Plant height and leaf area were assessed on five random plants monthly until 90 dat. Plant height was measured from the crown to the most apical point of the plant without spreading the leaves and expressed in cm. Leaf area was calculated by measuring the width and length of each leaf on the plant and expressed in cm^2^.

Aerial fresh weight, aerial dry weight, and plant coverage were assessed on three randomly selected plants at 90 dat and at harvest. Fresh weight was measured by removing the aerial part of the plant and weighing it on an Axis Aka 4200 high-precision balance. Aerial dry weight was determined by first drying the plants at room temperature in a hermetically sealed room for three weeks, followed by a drying process in a Yamato Scientific DS-64 oven at 70 °C for three days [27]. Plant coverage was assessed using the quadrat method, placing a 50 × 50 cm square at the base of each plant, subdivided into 25 squares of 10 × 10 cm, and counting the quadrats covered with vegetation.

### 2.6. Root Staining and Quantification of Mycorrhizal Colonization

For the observation of arbuscular mycorrhizae, the protocol described by Koske and Gemma [28] was followed. Root samples were taken in triplicate from each treatment and rinsed thoroughly with water to remove soil residues. The youngest and finest roots (preferably lateral roots) were then cut into approximately 1 cm pieces and placed in a bottle with 50 mL of 2.5% (*w*/*v*) KOH to decolorize them. The samples were incubated in a water bath at 90 °C for 1.5 h. After this time, the KOH solution was discarded, the roots were rinsed with water, and 50 mL of 1% HCl was added, allowing the roots to incubate overnight. The acidified roots were then stained with 0.05% trypan blue in acidified glycerol and incubated in a water bath at 90 °C for 1 h. Mycorrhizal colonization was quantified according to the methodology described by Giovanetti and Mosse [29]. The stained root segments were placed on a Petri dish with a grid drawn at the reverse. The number of intersections between grid lines and total roots (TRI) and the number of grid intersections with mycorrhizal roots (MRI) were recorded. The percentage of mycorrhizal colonization (%MC) was calculated using the following formula:$$\%\;MC=\left(\frac{MRI}{TRI}\right)\times 100$$

### 2.7. Relative Agronomic Efficiency Percentage (RAE%)

The relative agronomic efficiency percentage (RAE%) for plant height and leaf area at 90 dat was calculated using the following formula proposed by León et al. [30]:% RAE=((A−B)/B)×100

This expresses the percentage increase in height (cm) and leaf area (cm^2^) of the experimental treatments concerning the height and leaf area of the control.

Where A is the plant height (cm) or leaf area (cm^2^) of the experimental treatment, and B is the plant height (cm) or leaf area (cm^2^) of the control treatment without inoculation.

### 2.8. Yield Components

The number of fruits per m^2^ was determined by manually counting all fruits per experimental area (4.8 m^2^) of each replicate. Counting was carried out weekly, between 90 and 160 dat. The number of fruits accumulated up to the time of harvest was calculated.

To determine fruit weight, five plants were randomly selected in each experimental plot. The individual fruit weights of all fruits from these five plants were measured weekly between 90 and 160 dat by using an Axis Aka 4200 high-precision balance. The final result was calculated as the overall average of all these measurements.

Yield (t∙ha^−1^) was determined by estimating the average individual fruit weight and the number of fruits per m^2^.

### 2.9. Fruit Quality Components

At harvest time, at 160 dat, 10 fruits per replicate were collected. The following analyses were performed: pH was evaluated with a Hanna HI2020 multiparameter meter (AOAC method 981.12); acidity by titration with 0.1 N sodium hydroxide (AOAC method 942.15); Brix degrees by using a PAL-1 Atago 3810 digital refractometer; and firmness with a Lutron FR-5120 penetrometer.

### 2.10. Statistical Analysis

The obtained data in the different mentioned parameters were analyzed by using ANOVA with a significance level of 0.05, after confirming the normality of the data and the homogeneity of variances. Mean comparisons were performed using the least significant difference (LSD Fisher) test. The statistical analysis and Principal Component analysis were conducted using R software version 4.3.1 (Lucent Technologies, Murray Hill, NJ, USA).

## 3. Results

### 3.1. Mycorrhizal Colonization

It was observed that inoculated treatments with *Rhizophagus intraradices* exhibited a statistically higher percentage of mycorrhizal colonization compared to both the non-inoculated treatment and the *Bacillus subtilis* inoculated treatment. This result was consistent regardless of whether chemical fertilization was present or absent (Figure 1).

### 3.2. Plant Height and Leaf Area

The results for both parameters indicated that microbial inoculation effects varied depending on the presence or absence of chemical fertilization. In the absence of chemical fertilization, although plants inoculated with *Bacillus subtilis* were taller at 60 dat, non-inoculated plants surpassed them by the end of the vegetative stage (90 dat). Conversely, in the presence of chemical fertilization, the *Bacillus subtilis* treatment consistently exhibited the highest height values at both 60 and 90 dat. At medium fertilization levels, non-inoculated plants even had the lowest height values. While no statistical significance was observed at the F50 and F100 levels, there is a tendency for *Bacillus subtilis* to synergize with chemical fertilization in enhancing plant height (Figure 2).

A similar trend was observed for the leaf area at the end of the vegetative stage (90 dat). Without fertilization, the non-inoculated treatment was statistically superior. However, with fertilization, the *Bacillus subtilis* treatment achieved the highest leaf area values, and at medium fertilization levels, it even statistically differed from the control (Figure 3).

Figure 4 illustrates the described trends more clearly. A positive relative agronomic efficiency percentage (RAE%) indicates that the inoculated treatment had, on average, higher values compared to the non-inoculated control. The RAE% calculations for plant height and leaf area reveal that the values for the *Bacillus subtilis* treatment are positive in the presence of chemical fertilization, compared to the non-inoculated treatment under chemical fertilization. Additionally, for plant height, both the *Rhizophagus intraradices* and co-inoculation treatments also exhibit positive RAE% values under medium fertilization.

### 3.3. Fresh and Dry Aerial Weight

Table 3 indicates that at harvest time, there were no significant differences in aerial fresh and dry weight among the treatments across any fertilization and inoculation scenarios. However, differences were observed at the end of the vegetative stage (90 dat). For aerial fresh weight, the non-inoculated treatment was statistically superior in the absence of fertilization. With medium and complete fertilization, the highest values were observed with *Bacillus subtilis* inoculation, although these differences were not statistically significant. The most notable aerial dry weight increase at 90 dat was achieved with the combination of complete fertilization (225-100-250 kg∙ha^−1^ of N, P_2_O_5_, and K_2_O) and *Bacillus subtilis* inoculation. This treatment resulted in an aerial dry weight of 12.9 ± 1.9 g, compared to 5.9 ± 1.7 g for the complete fertilization treatment without inoculation. This represents an increase of 144% to 352% in the fertilization effect due to inoculation.

### 3.4. Plant Coverage

Plant coverage is the proportion of area occupied by the vertical projection towards the soil, from the aerial parts of a plant. As previously described, it was observed that in the absence of chemical fertilization, the non-inoculated treatment was statistically superior to the inoculated ones. With medium fertilization, the inoculated treatments showed positive and higher RAE% values than the non-inoculated control, although the differences were not statistically significant. However, with complete fertilization, the *Bacillus subtilis* treatment was statistically superior to the non-inoculated treatment, resulting in a 50% increase in plant coverage (Figure 5).

### 3.5. Fruit Quality

Table 4 presents the results for pH, titratable acidity, Brix degrees, and fruit firmness. Inoculation with *Bacillus subtilis*, either individually or in combination with *Rhizophagus intraradices*, significantly increased the maturity index (Brix: acidity) by 64% and fruit firmness by 68% compared to non-inoculated plants. These improvements were consistent regardless of the presence or absence of chemical fertilization. The treatment with average fertilization doses of 112-50-125 kg∙ha^−1^ of N, P_2_O_5_, and K_2_O, combined with *Bacillus subtilis* inoculation, achieved the best fruit quality characteristics, with a soluble sugar to titratable acidity ratio of 27 and fruit firmness of 0.181 ± 0.02 kg.

### 3.6. Yield

Table 5 reveals that significant differences in the number of fruits per m^2^ and yield in t∙ha^−1^ were observed only with complete fertilization at 225-100-250 kg∙ha^−1^ of N, P_2_O_5_, and K_2_O. The *Bacillus subtilis* inoculated treatment achieved the highest number of fruits per m^2^ and the highest yield in t∙ha^−1^, with increases of 46.8% and 46.9%, respectively, compared to the treatment with complete fertilization without inoculation.

Productive efficiency determination of the plant, based on yield per gram of leaf dry matter, showed that the combination of complete fertilization at 225-100-250 kg∙ha^−1^ of N, P_2_O_5_, and K_2_O with *Bacillus subtilis* inoculation achieved a 215.60 g production per gram of dry matter. This represents a 56.93% increase compared to the treatment with complete fertilization without inoculation.

For the medium fertilization treatment of 112-50-125 kg∙ha^−1^ of N, P_2_O_5_, and K_2_O, significant differences were observed in yield efficiency determination relative to leaf area (cm^2^) and leaf dry matter (g). The combination of medium fertilization with *Rhizophagus intraradices* inoculation, both individually and in consortium, achieved the highest yield efficiency per cm^2^ of leaf area and gram of leaf dry matter (Table 5).

### 3.7. Principal Component Analysis

Principal component analysis (PCA) confirmed the observed patterns in our results. In the absence of fertilization, the non-inoculated treatment excelled in most vegetative variables (Figure 6A), as depicted in the positive sections of components 1 and 2 (Figure 6B). With medium fertilization, the *Bacillus subtilis* treatment achieved the highest values in vegetative variables, while *Rhizophagus intraradices* (both individually and in co-inoculation) excelled in yield variables (Figure 6C), as indicated in the negative section of component 1 and the positive section of component 2 (Figure 6D). When complete fertilization was applied, the *Bacillus subtilis* treatment demonstrated superior values across vegetative, yield, and fruit quality variables, whereas the control treatment was positioned opposite in all these aspects (Figure 6E,F).

### 3.8. Heatmap Graph Analysis

The heatmap analysis (Figure 7) provided a comprehensive view of all treatments under different fertilization scenarios. It was evident that microbial inoculation positively influenced the strawberry crop, particularly enhancing yield and fruit quality compared to non-inoculated treatments. The best results were achieved under *Bacillus subtilis* inoculation (cluster C), especially when combined with fertilization (50% and 100%), as it led to significant improvements in both productive and biometric parameters. In Cluster C, it was also observed that the treatment without fertilization and without inoculation showed high values in the vegetative stage; however, it did not stand out in the productive stage. *Rhizophagus intraradices* did not show a positive influence on the vegetative stage of the strawberry. However, it did have good results in the yield indexes, but only when it was found next to a 50% fertilization (Cluster A). Co-inoculation also did not show positive effects in the vegetative stage, but did show good results in fruit quality and yield index, independent of the fertilization scenarios (Cluster B), demonstrating its versatility.

## 4. Discussion

There are several studies that demonstrate microbial inoculation improves nutrient use efficiency [31,32]. In our study, it was observed that plants inoculated with *Bacillus subtilis*, in the presence of mineral fertilization, showed an increase in biometric parameters, such as height (Figure 2), leaf area (Figure 3), plant coverage (Figure 5), and aerial dry weight (Table 3). This could reaffirm that *B. subtilis* improved nutrient use; however, foliar analyses are needed to verify this. The most effective treatment for stimulating increases in leaf area and plant height was the combination of an average fertilizer dose of 112-50-125 kg·ha^−1^ (half dose) of N, P_2_O^5^, and K_2_O, along with *Bacillus subtilis* inoculation. These findings align with the results reported by Bueno et al. [33], who observed that soybean plants inoculated with *Bacillus subtilis* at various concentrations (1 × 10^2^ to 1 × 10^10^ cfu·mL^−1^) showed optimal height growth when fertilized with doses between 50% and 60% of the recommended dose. Additionally, Agbodjato et al. [34] found that the tallest maize plants were achieved with microbial inoculation and fertilization at 50% of the recommended N, P_2_O_5_, and K_2_O levels. Bueno et al. [33] also noted that while microbial inoculation with *Bacillus subtilis* increases the plant’s fertilization needs, the results of the present study indicate lower plant height, leaf area, fresh weight, and aerial dry weight in inoculated strawberry crops that did not receive fertilization.

Regarding aerial biomass, Bueno et al. [33] reported that the highest biomass production was achieved with 52% fertilization and a 1 × 10^7^ cfu·mL^−1^ inoculum concentration. In our study, the treatment involving *Bacillus subtilis* and fertilization doses of 225-100-250 kg·ha^−1^ (full dose) of N, P_2_O_5_, and K_2_O resulted in the highest dry matter production capacity, with statistically significant differences observed (Table 3). Additionally, this treatment also enhanced plant coverage (Figure 5), with *Bacillus subtilis* inoculation proving superior to the non-inoculated treatment when complete fertilization was applied.

The *Bacillus* genus is commonly found in the rhizosphere, and specifically, *Bacillus subtilis* is frequently reported as a Plant-Growth Promoting Rhizobacterium (PGPR) across various crops. This is attributed to its capacity to produce beneficial metabolites such as cytokinins, siderophores, auxins, antibiotics, organic acids, and VOCs [35]; solubilize phosphates; confer stress tolerance to the plant; induce systemic resistance; and form biofilms, among other functions [35]. Regarding the strawberry context, several studies have investigated the *Bacillus* species’ potential as biocontrol agents against pathogens like *Botrytis cinerea* [36,37], *Fusarium* spp. [38,39], *Verticillium* spp. [39], and *Colletotrichum* spp. [40]. However, fewer studies have evaluated *Bacillus* spp. as growth promoters for strawberries under field conditions [41]. In contrast, research on other crops has demonstrated that *Bacillus subtilis* has a positive impact on vegetative parameters, which supports our findings. For instance, Lima et al. [42] reported that *Bacillus subtilis* PRBS-1 enhanced aerial and root biomass in maize plants subjected to abiotic stress, while Braga et al. [43] observed increased biomass and productivity in soybean plants under field conditions following inoculation with *Bacillus subtilis* UFT-Bs10. In order to understand the mechanisms by which *B. subtilis* improved growth in strawberry seedlings, it is necessary to carry out physiological studies.

On the other hand, the overall effect of *Rhizophagus intraradices* inoculation on vegetative parameters was low and was not significantly favored by the presence of fertilizer (Figure 7). These results do not agree with Oliveira et al. [44], where phosphorus fertilization did affect mycorrhizal colonization in *Schinopsis brasiliensis*, decreasing its effect on biomass production. Conversely, Mitova et al. [45] observed in their study of lettuce inoculated with *Glomus intraradices* that nitrogen fertilization had the most favorable impact on plant biomass. In this research, mycorrhizae effects on vegetative parameters were low in all fertilization scenarios. It is likely that in this study, the low effect of mycorrhizae is due more to the carrier in which the mycorrhizae were found and less to the presence or absence of fertilization. The inoculum with the *Rhizophagus intraradices* strain used in the experiment had biochar and sand as a carrier. It has been reported that the combination of mycorrhizae with organic substrates does not always have positive effects. For example, Püschel et al. [46] mentioned that the decrease in mycorrhizal infectivity is attributed to unfavorable properties inherent to peat, while Sun et al. [47] observed that the presence of biochar significantly reduces mycorrhizal colonization, and Barna et al. [48] found that a higher concentration of biochar decreased glomalin production. This indicates the need for further investigation into mycorrhizal inoculation in strawberry cultivation, as its effects can vary depending on fertilization practices and its interactions with other amendments.

Flavor is considered a crucial aspect of fruit quality in strawberries [49]. Although it is challenging to measure and quantify, the ripeness index (TSS/TA ratio) is commonly used as an indicator of sweetness and overall consumer acceptance [50]. Additionally, strawberries are highly perishable [49], prompting researchers to seek methods for extending their post-harvest shelf life. One major factor contributing to their short shelf life is their susceptibility to mechanical damage during transport and storage. Enhancing fruit firmness is critical for reducing post-harvest losses in strawberries [51]. In our study, inoculation with *Bacillus subtilis* led to a ~64% increase in the fruit maturity index and a ~56% increase in fruit firmness (Table 4), both of which are critical factors for strawberry marketing. Notably, these effects were independent of mineral nutrition. Similar improvements have been observed in other fruits; for instance, Abraham-Juárez et al. [52] reported that *Bacillus subtilis* (BEB-23, BEB-22, and BEB-13) strains increased firmness and ripening index in melon fruits compared to non-inoculated controls. Additionally, Qiu et al. [53] found that inoculation with a commercial *Bacillus subtilis*-based product increased the ripening index of oranges by 9 to 21%.

Growth-promoting microorganisms can make a greater amount of nutrients available to the plant, which can be reflected in an increase in crop yield [54]. In our study, inoculation with *Bacillus subtilis* increased strawberry yield by 46.9% (amount of fruits harvested per square meter and tons per hectare) when the plants were fertilized at full dose, compared to non-inoculated plants (Table 5). These results are similar to those obtained by Chebotar et al. [41] in their comparative study of nitrogen fertilizers with inoculation of *Bacillus velezensis* BS89. The authors concluded that the application of this microorganism increased the productivity of two strawberry varieties either in the absence or presence of nitrogen fertilizers.

Microbial inoculants indeed emerge as an alternative to the excessive use of chemical fertilizers. However, it is important to highlight that these products cannot yet be completely dispensed with, since it has been observed that their application together with microbial inoculation influences positively the crop. Various studies conclude that the appropriate use of chemical and biological inputs can increase crop yields [41,55].

## 5. Conclusions

The results showed different behaviors of microbial inoculation in the presence and absence of mineral fertilization. This shows us that more studies should be carried out on factors that can influence the efficiency of microbial inoculation in crops.

Inoculation with *Bacillus subtilis* at a concentration of 10^7^ cfu·mL^−1^, supplemented with 100% mineral fertilization (225 kg·ha^−1^ of N, 100 kg·ha^−1^ of P_2_O_5_ and 250 kg·ha^−1^ of K_2_O), achieved the best results in terms of biometric variables, yield, and fruit quality components.

The study found that the biostimulant effect of *Bacillus subtilis* as a plant growth-promoting rhizobacteria probably increased the nutritional demands of strawberry plants, since treatments that involved only microbial inoculation, without adequate fertilization, led to a decrease in biometric characteristics and crop production. On the other hand, when *Bacillus subtilis* was used together with complete fertilization, improvements in vegetative state and yield were observed.

Under the conditions in which the experiment was carried out, these results demonstrate the synergistic effect of combining fertilization with the inoculation of plant growth-promoting microorganisms. However, it is important to repeat this treatment in other seasons and regions to validate the effect observed.

## Figures and Tables

**Figure 1 microorganisms-12-01816-f001:**
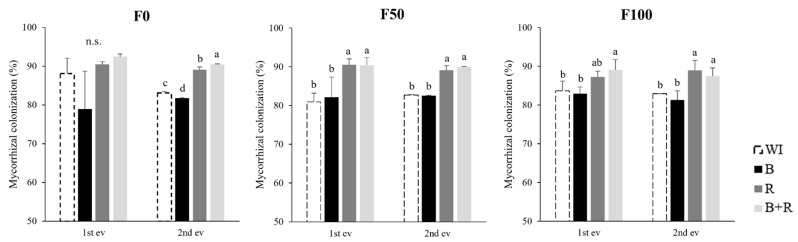
Mycorrhizal colonization (%) based on microbial inoculation evaluated at the end of the vegetative stage (1st evaluation, 90 dat) and at harvest (2nd evaluation, 160 dat), at different fertilization levels (F0 = without fertilization, F50 = 112-50-125 kg·ha^−1^ of N, P_2_O_5_, K_2_O, F100 = 225-100-250 kg·ha^−1^ of N, P_2_O_5_, K_2_O, WI = without inoculation, B = inoculation with *Bacillus subtilis*, R = inoculation with *Rhizophagus intraradices*, B + R = co-inoculation). Different letters for the same fertilization level are statistically different (LSD Fisher, α = 0.05). n.s. no significance.

**Figure 2 microorganisms-12-01816-f002:**
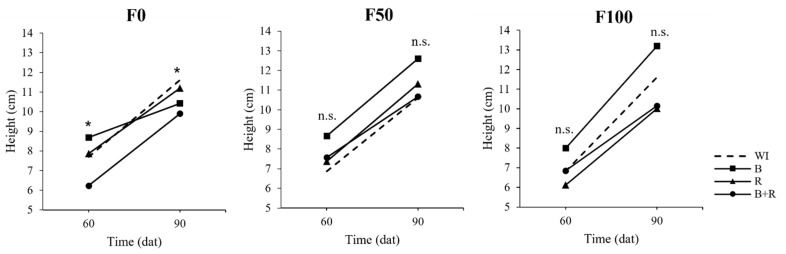
Plant height based on microbial inoculation at 60 and 90 dat in strawberry plants cv. San Andreas at different fertilization levels (F0 = without fertilization, F50 = 112-50-125 kg·ha^−1^ of N, P_2_O_5_, K_2_O, F100 = 225-100-250 kg·ha^−1^ of N, P_2_O_5_, K_2_O, WI = without inoculation, B = inoculation with *Bacillus subtilis*, R = inoculation with *Rhizophagus intraradices*, B + R = co-inoculation). * Statistical significance according to ANOVA (α = 0.05), n.s. no significance.

**Figure 3 microorganisms-12-01816-f003:**
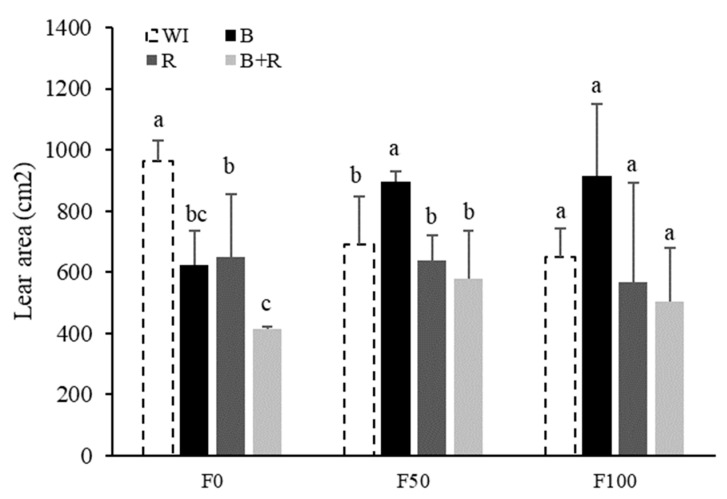
Leaf area (cm^2^) based on microbial inoculation at 90 dat in strawberry plants cv. San Andreas at different fertilization levels (F0 = without fertilization, F50 = 112-50-125 kg·ha^−1^ of N, P_2_O_5_, K_2_O, F100 = 225-100-250 kg·ha^−1^ of N, P_2_O_5_, K_2_O, WI = without inoculation, B = inoculation with *Bacillus subtilis*, R = inoculation with *Rhizophagus intraradices*, B + R = co-inoculation). Different letters in the same fertilization level are statistically different (LSD Fisher, α = 0.05).

**Figure 4 microorganisms-12-01816-f004:**
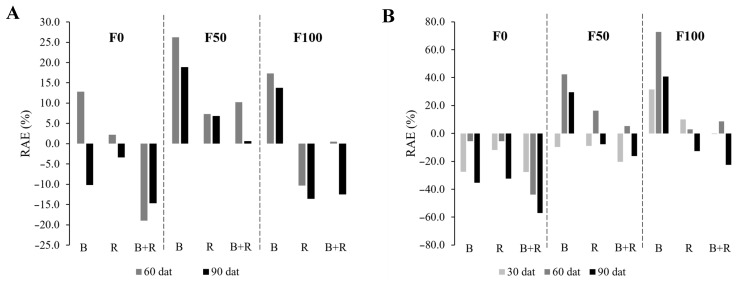
Relative Agronomic Efficiency (RAE %) of plant height (**A**) and leaf area (**B**) based on microbial inoculation at different fertilization levels (F0 = without fertilization, F50 = 112-50-125 kg·ha^−1^ of N, P_2_O_5_, K_2_O, F100 = 225-100-250 kg·ha^−1^ of N, P_2_O_5_, K_2_O, WI = without inoculation, B = inoculation with *Bacillus subtilis*, R = inoculation with *Rhizophagus intraradices*, B + R = co-inoculation).

**Figure 5 microorganisms-12-01816-f005:**
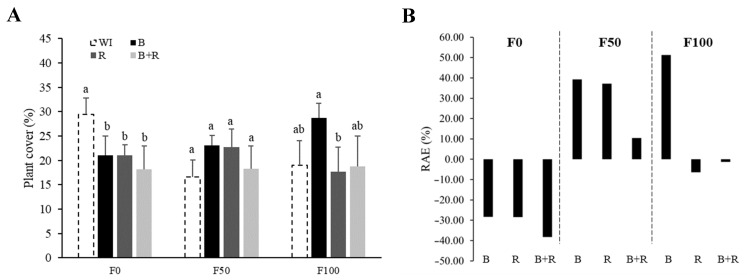
Plant coverage (%) based on microbial inoculation evaluated at the end of the vegetative stage (90 dat) at different fertilization levels (**A**) and RAE% for plant cover (**B**) (F0 = without fertilization, F50 = 112-50-125 kg·ha^−1^ of N, P_2_O_5_, K_2_O, F100 = 225-100-250 kg·ha^−1^ of N, P_2_O_5_, K_2_O, WI = without inoculation, B = inoculation with *Bacillus subtilis*, R = inoculation with *Rhizophagus intraradices*, B + R = co-inoculation). Different letters for the same fertilization level are statistically different (LSD Fisher, α = 0.05).

**Figure 6 microorganisms-12-01816-f006:**
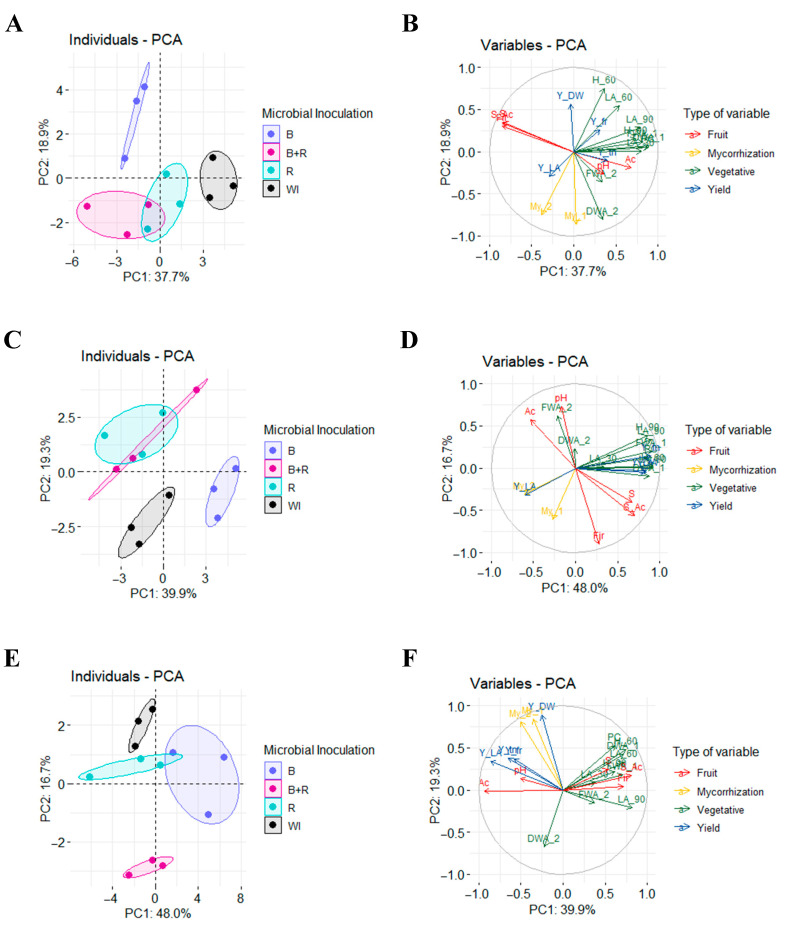
Principal Component Analysis (PCA) of evaluated parameters according to microbial inoculation without fertilization (**A**,**B**), with half dose of fertilization (**C**,**D**) and with full dose of fertilization (**E**,**F**). (**A**,**C**,**E**) Distribution of inoculation treatments in the principal components (WI = without inoculation, B = inoculation with *Bacillus subtilis*, R = inoculation with *Rhizophagus intraradices*, B + R = co-inoculation). (**B**,**D**,**F**) Distribution of the variables in the principal components (LA = leaf area; H = height; FWA = fresh weight aerial; DWA = dry weight aerial; PC = plant cover; My = mycorrhization; pH = pH; Ac = acidity; Fir = firmness; S = sugars; S_Ac = ratio sugars: acidity; Y_fr = yield expressed in fruit per square meter; Y_tn = yield expressed in tons per hectare; Y_LA = yield: leaf area index; Y_DW = yield: dry weight index; _30 = at 30 dat; _60 = at 60 dat; _90 = at 90 dat; _1 = first evaluation; _2 = second evaluation).

**Figure 7 microorganisms-12-01816-f007:**
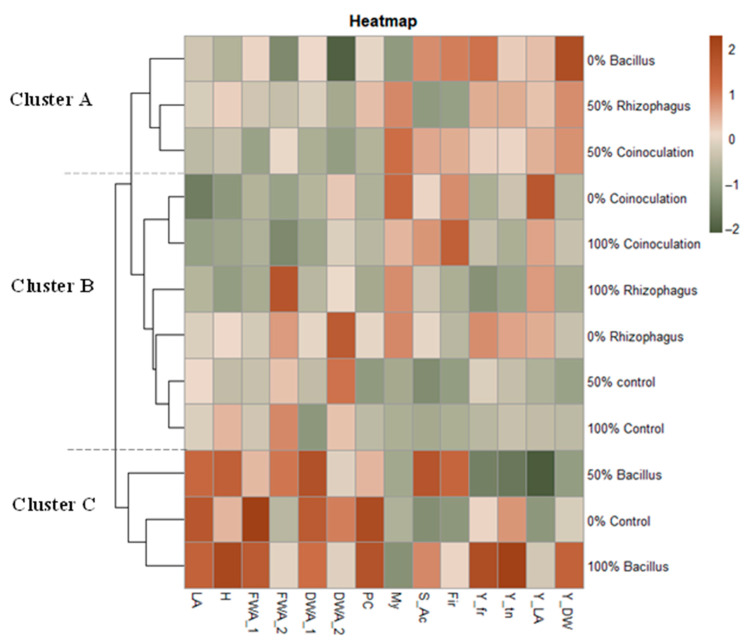
Heatmap with cluster analysis of the combined effects between chemical fertilization and microbial inoculation. The *Y*-axis indicates the doses of chemical fertilization and the inoculated microorganism. The *X*-axis indicates the evaluated parameters (LA = leaf area; H = height; FWA = fresh weight aerial; DWA = dry weight aerial; PC = plant cover; My = mycorrhization; S_Ac = ratio sugars: acidity; Fir = firmness; Y_fr = yield expressed in fruit per square meter; Y_tn = yield expressed in tons per hectare; Y_LA = yield: leaf area index; Y_DW = yield: dry weight index; _1 = first evaluation; _2 = second evaluation).

**Table 1 microorganisms-12-01816-t001:** Evaluated factors and treatment.

Treat.	Factor 1. Fertilization	Factor 2. Microbial Inoculation
1	Without fertilization (F0)	Without inoculation (WI)
2		*Bacillus subtilis* (B)
3		*Rhizophagus intraradices* (R)
4		Co-inoculation (B + R)
5	Half dose of fertilization (F50)	Without inoculation (WI)
6		*Bacillus subtilis* (B)
7		*Rhizophagus intraradices* (R)
8		Co-inoculation (B + R)
9	Full dose of fertilization (F100)	Without inoculation (WI)
10		*Bacillus subtilis* (B)
11		*Rhizophagus intraradices* (R)
12		Co-inoculation (B + R)

**Table 2 microorganisms-12-01816-t002:** Fertilization scheme.

Fertilizers	N° Fertilization				Total (g 4.8 m^−2^)
	1°	2°	3°	4°	
*Full dose (F100)*					
Urea (N)	58.18	45.25	45.25	45.25	193.93
Diammonium phosphate (P)	62.61	13.91	13.91	13.91	104.35
Potassium sulfate (K)	72	56	56	56	240.00
*Half dose (F50*)					
Urea (N)	29.09	22.62	22.62	22.62	96.97
Diammonium phosphate (P)	31.31	6.95	6.95	6.95	52.18
Potassium sulfate (K)	36.00	28.00	28.00	28.00	120.00

**Table 3 microorganisms-12-01816-t003:** Fresh and dry weight of the aerial part based on microbial inoculation at 90 dat and harvest, and RAE% for both parameters.

Fert.	Inoc.	Fresh Weight		RAE% *	Dry Weight		RAE% *
		90 dat	Harvest		90 dat	Harvest	
F0	WI	61.0 ± 10.3 ^a^	91.1 ± 26.4 n.s.		14.2 ± 2.0 ^a^	39.6 ± 8.2 n.s.	
	B	40.4 ± 2.6 ^b^	78.1 ± 15.0	−14.3	9.7 ± 0.3 ^b^	24.0 ± 6.3	−39.3
	R	36.6 ± 3.5 ^b^	112.9 ± 13.9	**23.9**	9.3 ± 1.0 ^b^	42.7 ± 8.9	**7.8**
	B + R	32.5 ± 9.2 ^b^	84.5 ± 23.7	−7.3	7.5 ± 2.1 ^b^	36.0 ± 10.1	−8.9
F50	WI	34.9 ± 7.9 n.s.	106.3 ± 21.0 n.s.		7.9 ± 2.3n.s.	40.2 ± 7.0 n.s.	
	B	43.1 ± 11.8	118.7 ± 22.1	**11.6**	14.8 ± 6.4	33.8 ± 10.5	−15.9
	R	36.0 ± 5.4	93.3 ± 30.4	−12.3	8.9 ± 1.5	30.0 ± 7.5	−25.5
	B + R	29.5 ± 12.5	101.3 ± 34.2	−4.7	7.1 ± 0.7	28.8 ± 6.5	−28.4
F100	WI	36.0 ± 11.1 n.s.	115.6 ± 32.6 n.s.		5.9 ± 1.7 ^b^	36.1 ± 8.0 n.s.	
	B	54.5 ± 8.8	99.3 ± 47.5	−14	12.9 ± 1.9 ^a^	33.8 ± 3.1	−6.4
	R	31.2 ± 12.7	129.6 ± 43.7	**12.1**	7.7 ± 3.1 ^b^	34.7 ± 5.5	−3.9
	B + R	32.2 ± 14.1	77.8 ± 10.2	−32.6	6.6 ± 0.7 ^b^	33.6 ± 12.0	−7

* RAE% in bold = value higher than the control, F0 = without fertilization, F50 = half dose, F100 = full dose, WI = without inoculation, B = inoculation with *Bacillus subtilis*, R = inoculation with *Rhizophagus intraradices*, B + R = co-inoculation. Means with different letters in the same column for each fertilization level are statistically different (LSD Fisher, α = 0.05). n.s. no significance.

**Table 4 microorganisms-12-01816-t004:** Physical–chemical characteristics of fruit based on inoculation at different fertilization levels.

Fert.	Inoc.	pH	Acidity	Sugars	Ratio	Firmness
			g. citric acid/100 mL	Brix°	Brix: Acidity	kg
F0	WI	3.32 ± 0.11 n.s.	0.77 ± 0.00 ^a^	11.2 ± 0.3 ^c^	14.6 ^c^	0.100 ± 0.01 ^c^
	B	3.28 ± 0.05	0.62 ± 0.04 ^b^	14.4 ± 0.5 ^a^	23.4 ^a^	0.169 ± 0.01 ^a^
	R	3.36 ± 0.07	0.64 ± 0.06 ^b^	12.6 ± 1.3 ^bc^	19.7 ^b^	0.119 ± 0.01 ^b^
	B + R	3.32 ± 0.11	0.69 ± 0.02 ^ab^	14.1 ± 0.8 ^ab^	20.4 ^ab^	0.166 ± 0.01 ^a^
F50	WI	3.36 ± 0.06 n.s.	0.72 ± 0.02 ^a^	10.2 ± 0.4 ^c^	14.3 ^c^	0.104 ± 0.00 ^c^
	B	3.09 ± 0.03	0.58 ± 0.00 ^b^	15.5 ± 0.4 ^a^	27.0 ^a^	0.181 ± 0.02 ^a^
	R	3.42 ± 0.01	0.78 ± 0.08 ^a^	11.5 ± 0.6 ^b^	15.3 ^c^	0.104 ± 0.01 ^c^
	B + R	3.31 ± 0.09	0.69 ± 0.10 ^ab^	15.1 ± 0.3 ^a^	22.2 ^b^	0.156 ± 0.01 ^b^
F100	WI	3.37 ± 0.12 n.s.	0.70 ± 0.03 n.s.	11.5 ± 0.2 ^c^	16.3 ^b^	0.113 ± 0.01 ^c^
	B	3.29 ± 0.08	0.66 ± 0.04	15.5 ± 0.5 ^a^	23.5 ^a^	0.143 ± 0.02 ^b^
	R	3.38 ± 0.08	0.69 ± 0.07	12.8 ± 0.2 ^bc^	18.5 ^b^	0.114 ± 0.01 ^c^
	B + R	3.22 ± 0.01	0.62 ± 0.04	14.2 ± 1.6 ^ab^	23.0 ^a^	0.184 ± 0.00 ^a^

F0 = without fertilization, F50 = half dose, F100 = full dose, WI = without inoculation, B = inoculation with *Bacillus subtilis*, R = inoculation with *Rhizophagus intraradices*, B + R = co-inoculation. Means with different letters in the same column for each fertilization level are statistically different (LSD Fisher, α = 0.05). n.s. no significance.

**Table 5 microorganisms-12-01816-t005:** Yield and ratios based on leaf area and dry weight aerial of the strawberry crop.

Fert.	Inoc.	Yield		Yield:LA	Yield:DW
		Fruits per m^2^	t·ha^−1^		
F0	WI	21.8 ± 6.0 n.s.	5.9 ± 0.9 n.s.	6.1 ± 1.1 n.s.	151 ± 28 n.s.
	B	25.3 ± 3.2	5.4 ± 2.2	9.3 ± 4.8	233 ± 117
	R	24.3 ± 9.3	5.8 ± 1.9	9.7 ± 5.4	143 ± 64
	B + R	18.6 ± 5.8	4.9 ± 1.5	11.8 ± 3.7	136 ± 28
F50	WI	20.7 ± 4.4 n.s.	4.8 ± 1.0 n.s.	7.1 ± 1.6 ^b^	120 ± 18 ^b^
	B	15.9 ± 1.8	3.8 ± 0.6	4.2 ± 0.7 ^c^	116 ± 21 ^b^
	R	23.2 ± 5.2	5.7 ± 1.0	9.2 ± 3.0 ^ab^	192 ± 12 ^a^
	B + R	21.9 ± 4.1	5.4 ± 1.0	9.5 ± 1.8 ^a^	191 ± 42 ^a^
F100	WI	19.2 ± 1.3 ^b^	4.9 ± 0.6 ^b^	7.5 ± 0.3 n.s.	137 ± 17 ^b^
	B	28.2 ± 4.5 ^a^	7.2 ± 1.4 ^a^	7.9 ± 1.0	215 ± 60 ^a^
	R	16.8 ± 2.7 ^b^	4.4 ± 0.8 ^b^	9.9 ± 5.9	125 ± 10 ^b^
	B + R	19.7 ± 4.9 ^b^	4.5 ± 0.6 ^b^	9.8 ± 3.4	142 ± 32 ^b^

F0 = without fertilization, F50 = half dose, F100 = full dose, WI = without inoculation, B = inoculation with *Bacillus subtilis*, R = inoculation with *Rhizophagus intraradices*, B + R = co-inoculation. Means with different letters in the same column for each fertilization level are statistically different (LSD Fisher, α = 0.05). n.s. no significance.

## Data Availability

The data presented in this study are available on request from the corresponding author.

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
