# Peer review of "Bacillus subtilis and Rhizophagus intraradices Improve Vegetative Growth, Yield, and Fruit Quality of Fragaria × ananassa var. San Andreas"

_microorganisms, 2024, doi:10.3390/microorganisms12091816_

Round 1
Reviewer 1 Report
Comments and Suggestions for Authors
the work focusses on effects of mycorrhizal and Bacillus inocula in an alkaline soil on strawberry : effects of NPK additions are considered
what is missing is analysis of the plant tissues- your claim that the Bs improves nutrition of the plant is not backed up by analysis - it is based on growth parameters which is OK
but could you address why other attributes known for Bs were not considered eg changes in essential metal availabity induced systemic stress tolerance etc ie the scope of discussion of beneficial Bs effects is limited
there also is little discussion about the mycorrhizal responses - was this just a pore choice of inoculum for efficacy under test conditions
there should have been an evaluation of colonization by the Bs inocula
plus looking at root morphology could have shed more mechanistic understanding
so the findings although sound an extensive remain at the observational level which does indeed allow speculation
comments are made as sticky notes- more details and references are needed in methods for instance

Reviewer 2 Report
Comments and Suggestions for Authors
The study entitled “Bacillus subtilis and Rhizophagus intraradices improve vegetative growth, yield, and fruit quality of Fragaria x ananassa var. San Andreas ”. evaluated the inoculation with Bacillus subtilis and/or Rhizophagus intraradices under different rates of chemical fertilization on the growth and productivity of strawberry plant. The subject is very interesting, and the methodology used is adequate for the objectives of the study. The results are of interest and support the conclusions. The manuscript is worth publishing in “Microorganisms”. However, there are still some issues that need to be addressed, I suggest major revision.
Specific comments
L35: Please use specific keywords and synonyms (Scientific names could not be as keywords). In addition, please arrange the keywords in alphabetical order.
L37: I suggest the author to start by the strawberry plant and its economic benefits, then extend to its problem with chemical fertilizers in general, then you can give the solution for the problem using microbial inoculation.
L43-44: This sentence needs to be clarified. What is the relationship between chemical fertilization and HM accumulation in the soil?
L49: What did the author exactly mean by concentration?
L52: What are other beneficial functions? I recommend extending the role of PGPRs in sustainable agriculture. This is far from the current knowledge. This paper will help you https://doi.org/10.3390/microorganisms12061123
L66-70: Please provide a solid hypothesis to give the reader more information regarding the purpose and the mechanistic used to achieve this goal, then may refer some lack in the previous study regarding some aspects (the novelty).
I would suggest updating the references in the introduction section. The main theme of this manuscript, Plant microbe interaction, is a hot topic. Therefore, many studies have been published since 2022 that could be referenced in this manuscript.
L115-116: Did the authors use any adhesive coagent?
L119: Scientific names should be italic in the whole manuscript.
L122-123: How many mycorrhizal spores are in 0.5g inoculum?
L124: Did the non-treated plants receive autoclaved mycorrhizal inoculum to get the same nutrients?
L154: Why was not the method of Vierheilig with ink instead of toxic trypanblue used? Vierheilig H., Coughlan A. P., Wyss U., Piché Y. (1998). Ink and vinegar, a simple staining technique for arbuscular-mycorrhizal fungi. Applied and Environmental Microbiology 64, 5004-5007.
L195-206: Mycorrhizal colonization.
L202: Superscript.
Figure 1: Please add the explanation for subfigures (A-C) in the figure. The same for all figures.
I suggest that the authors could add one sentence at the end of each paragraph to conclude the whole paragraph to make it easy for the reader.
I appreciate the well-written discussion; however, I suggest that the authors could cite the Tables and Figures in the discussion part to make the readers in touch with their results.
L404,409,415,429,431,439: Please follow the journal notation in citing references.
L450-471: The conclusions of the experiment did not give good reflection regarding the results. Solid conclusions about the obtained results should be given. Instead of repeating the results, the authors should answer the following Questions? What is the novelty of this work? How could this study add more benefits for strawberry cultivation? What are the future prospectives?
Regards.
Comments on the Quality of English LanguageMinor editing of English language required
Round 2
Reviewer 2 Report
Comments and Suggestions for Authors
This is the second time I have evaluated this manuscript. The authors addressed all my comments, and the manuscript has been noticeably improved. Many thanks for their contribution.
Comments on the Quality of English LanguageThe English language is understandable and correct. Only minor editorial and stylistic corrections are required.
Author Response
We sincerely thank the reviewer for taking the time to review our manuscript and provide constructive feedback to improve it.
Comments 1: The English language is understandable and correct. Only minor editorial and stylistic corrections are required.
Response 1: Thank you for your comment. English corrections were made as suggested by the academic editor.